# DRAMS: A tool to detect and re-align mixed-up samples for integrative studies of multi-omics data

Yi Jiang[1,2,3], Gina Giase[4], Kay Grennan[5], Annie W. Shieh[5], Yan Xia[1,5], Lide Han[3], Quan Wang[2,3], Qiang Wei[2,3], Rui Chen[2,3], Sihan Liu[1], Kevin P. White[6,7], Chao Chen[1,8]*, Bingshan Li[2,3]*, Chunyu Liu[1,5,9]*

**1** Center for Medical Genetics & Hunan Key Laboratory of Medical Genetics, School of Life Sciences, Central South University, Changsha, Hunan, China, **2** Department of Molecular Physiology & Biophysics, Vanderbilt University, Nashville, Tennessee, United States of America, **3** Vanderbilt Genetics Institute, Vanderbilt University Medical Center, Nashville, Tennessee, United States of America, **4** School of Public Health, University of Illinois at Chicago, Chicago, Illinois, United States of America, **5** Department of Psychiatry, SUNY Upstate Medical University, Syracuse, New York, United States of America, **6** Institute for Genomics and Systems Biology, Department of Human Genetics, University of Chicago, Chicago, Illinois, United States of America, **7** Tempus Labs Inc, Chicago, Illinois, United States of America, **8** National Clinical Research Center for Geriatric Disorders, Xiangya Hospital, Central South University, Changsha, Hunan, China, **9** School of Psychology, Shaanxi Normal University, Xi'an, Shaanxi, China

\* chenchao@sklmg.edu.cn (CC); bingshan.li@vanderbilt.edu (BL); LiuCh@upstate.edu (CL)

**Data Availability Statement:** The PsychENCODE BrainGVEX data being used to test the method was deposited in Synapse with restricted access, according to NIH policy for data sharing and

## Abstract

Studies of complex disorders benefit from integrative analyses of multiple omics data. Yet, sample mix-ups frequently occur in multi-omics studies, weakening statistical power and risking false findings. Accurately aligning sample information, genotype, and corresponding omics data is critical for integrative analyses. We developed DRAMS (https://github.com/Yi-Jiang/DRAMS) to Detect and Re-Align Mixed-up Samples to address the sample mix-up problem. It uses a logistic regression model followed by a modified topological sorting algorithm to identify the potential true IDs based on data relationships of multi-omics. According to tests using simulated data, the more types of omics data used or the smaller the proportion of mix-ups, the better that DRAMS performs. Applying DRAMS to real data from the PsychENCODE BrainGVEX project, we detected and corrected 201 (12.5% of total data generated) mix-ups. Of the 21 mix-ups involving errors of racial identity, DRAMS re-assigned all data to the correct racial group in the 1000 Genomes project. In doing so, quantitative trait loci (QTL) (FDR<0.01) increased by an average of 1.62-fold. The use of DRAMS in multi-omics studies will strengthen statistical power of the study and improve quality of the results. Even though very limited studies have multi-omics data in place, we expect such data will increase quickly with the needs of DRAMS.

## Author summary

Sample mix-up happens inevitably during sample collection, processing, and data management. It leads to reduced statistical power and sometimes false findings. It is of great importance to correct mixed-up samples before conducting any downstream analyses.

patient protection (https://www.synapse.org/#!Synapse:syn3270007/wiki/234785). Part of the raw data were available in http://resource.psychencode.org/.

**Funding:** This work was supported by National Natural Science Foundation of China [31571312 and 81401114 to C.C., 31871276 to C.L.], the National Key Plan for Scientific Research and Development of China [2016YFC1306000 to C.C.], and National Institutes of Health [U01 MH103340-01 and 1R01ES024988 to C.L.]. The funders had no role in study design, data collection and analysis, decision to publish, or preparation of the manuscript.

**Competing interests:** I have read the journal's policy and the authors of this manuscript have the following competing interests: Kevin P. White is employed by Tempus Labs Inc. The other authors have declared that no competing interests exist. This paper has no conflict of interests with Tempus' business.

We developed DRAMS to detect and re-align mixed-up samples in multi-omics studies. The basic idea of DRAMS is to align the data and labels for each sample leveraging the genetic information of multi-omics data. DRAMS corrects sample IDs following a two-step strategy. At first, it estimates pairwise genetic relatedness among all the data generated from all the individuals. Because the different data generated from the same individual should share the same genetics, we can group all the highly related data and consider that the data from one group have only one potential ID. Then, we used a "majority vote" strategy to infer the potential IDs for individuals in each group. Other information, such as matching of genetics-based and reported sexes, omics priororacy, etc., were also used to identify the potential IDs. It has been proved that DRAMS performs very well in both simulation and PsychENCODE BrainGVEX multi-omics data.

This is a *PLOS Computational Biology* Methods paper.

## Introduction

Investigation of complex traits and disorders can use multiple omics data to systematically explore regulatory networks and causal relationships. Sample mix-ups can occur in omics experiments during sample collection, handling, genotyping, and data management. As the number of datasets to be integrated increases, the likelihood of error also multiplies. Sample mix-ups reduce statistical power and generate false findings. Not only is the detection and re-alignment of errors in data identifications (IDs) critical to ensuring accurate findings in integrative studies, such corrections can increase statistical power thus the number of positive findings [1].

For multi-omics data, the sample re-alignment procedure can be generally divided into two steps: first, to estimate genetic relatedness among the data of different omics and group together all the data of the same individual; then, to assign potential IDs for each data group. It is well-known that genetic information from the same individual should be identical regardless of the omics from which it originated. Using genotype data as a mediator, data originated from the same individual can be grouped together.

Several tools have been developed to estimate genetic relatedness for multi-omics data in many different ways, such as genotype concordance [2, 3], correlation of different quantifications [1, 2], correlation of variant allele fractions [4], concordance of sequencing reads [4], *etc*. The various methods made it possible to compare different data types, such as DNA sequencing, RNA sequencing, SNP array, *etc*. However, these tools mainly focused on implementing the first step of sample re-alignment, that is, to estimate genetic relatedness among multi-omics data. None of the existing tools has a systematic solution on determining the potential IDs for each data.

It is certain that, after grouping the highly related data, some data ID can be empirically corrected based on the "majority vote" strategy [1, 2, 5]. However, without a systematic solution, it only works for small data of low-dimension, difficult to scale up, and the results lack statistics-based confidence. As multi-omics data of more data dimension are expected in the near future, it is much more challenging to identify the potential ID. Especially when more than one data are labeled by mistake for a single individual. Actually, taking full advantage of information from data of more omics types makes data ID correction more accurate, turning a challenge into an opportunity.

Here, we described DRAMS, a tool to Detect and Re-Align Mixed-up Samples that leverages sample relationships in multi-omics data by directly comparing genotype data. DRAMS also uses a logistic regression model followed by a modified topological sorting algorithm to systematically re-align misclassifications. This tool integrates sample relationships among different omics types, concordance rates of genetics-based sexes and reported sexes, etc. With this design, DRAMS can be applied to as many omics types as possible. Using both simulated data and real data, we proved the accuracy and power of DRAMS in studies involving multi-omics data.

## Results

### Design of DRAMS

The goal of DRAMS is to detect and re-align mix-ups based on the grounds that all omics data originating from the same individual should match genotypes. DRAMS operates on a two-step process: first, we ensure that all omics data of the same samples were grouped together by their genotypic relatedness; after that, we find out the potential true ID of each data group (**Fig 1**).

For the first step, we extracted highly related data pairs. To accomplish this, we called genotypes from each omics dataset and estimated genetic relatedness among these data. After that, we grouped the data according to the highly-related data pairs. A "group" was defined as a set of data, in which each data was highly related to at least one of the other data. Based on the grouping results, we can classify relationships of data into three types. The first type contains

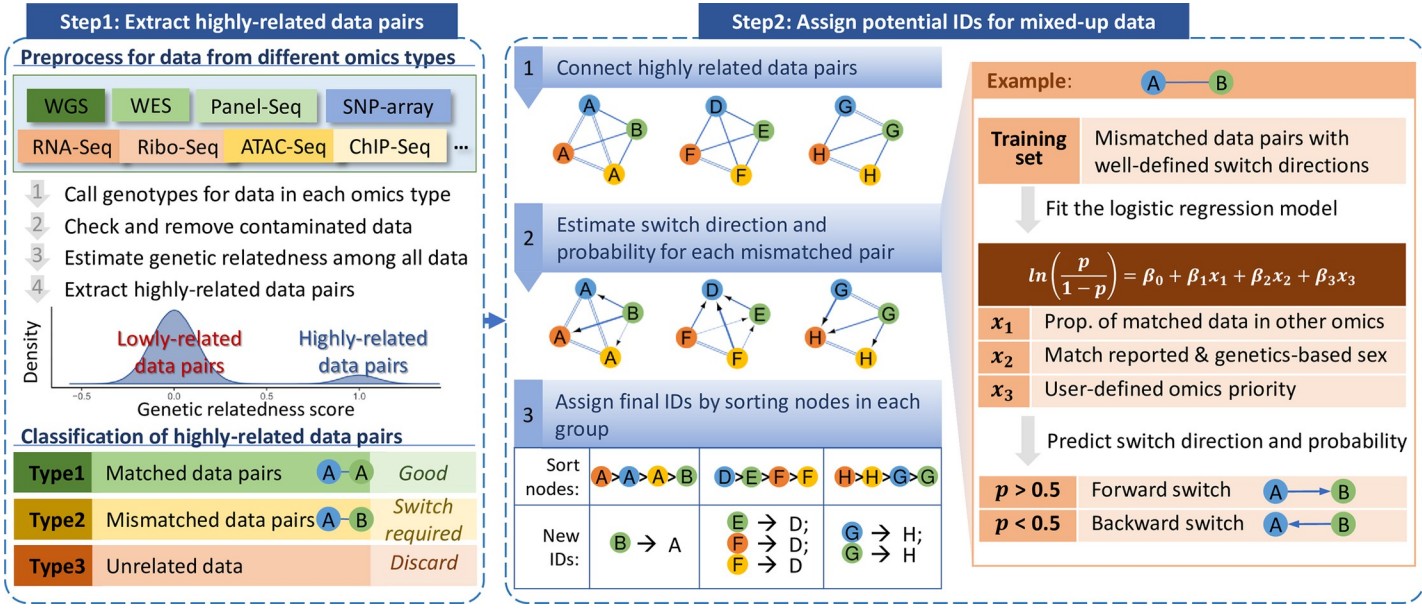

**Fig 1. Illustration of key steps.** The workflow has two steps: 1) extracting highly related data pairs; and 2) assigning potential data IDs. For the first step, genotypes were called for data from each omics types using GATK HaplotypeCaller. Contaminated data were checked and removed based on the VerifyBamID software and heterozygous rates. DRAMS then estimated genetic relatedness among all available omics data by comparing genotypes using GCTA. Any omics type that contains genetic information can be used here to call genotypes and compare genetic relatedness. Normally, the genetic relatedness scores for all data pairs were in bimodal distribution. Highly related data pairs were extracted based on the distribution of relatedness scores and connected to create multiple groups. Based on the matching of data IDs in each highly-related data pair, we can classify the highly-related data pairs as "matched data pairs" and "mismatched data pairs". We may also find some data unrelated to any other data. For the second step, we visualized the groups into multiple independent sub networks. For each sub network, each node represents a data in the group and each edge represents a highly-related data pair. The text in each node represents the data ID. Different colors represent different omics types. The parallel line connects matched data; whereas, the singular line connects mismatched data. After applying a logistic regression model, DRAMS tool estimated switch directions and probabilities for each mismatched data pair. The arrow denotes possible switch direction. The thickness of the line weight correlates to the degree of switch probability. The final IDs for the data in each group can be determined by sorting the nodes.

highly related data pairs that have the same individual IDs (**matched pairs**). These are the least likely to be mis-assigned. The second type contains highly related data pairs with different individual IDs (**mismatched pairs**), in which some individual IDs may have been swapped. We also have some data that are unrelated to any other. We put them into the third type, which will be discarded since they are unassignable.

For the second step, we connect all the highly related data pairs and produce multiple independent groups. Each node within a group represents one data point with each edge connecting a highly related data pair. Then, we use a multi-step, combined knowledge-based and statistical approach to search for the potential IDs in each of the groups, which contains both matched and mismatched data pairs. To estimate which data ID from the mismatched data pair was more likely a correct ID, we first use a logistic regression model. The model estimates the direction and probability based on three pieces of information: 1) data relationships among multiple omics types–data matching a greater number of omics sets are more likely to represent the true ID than those matching only a few; 2) when the reported sex of the data agrees with the genetics-based sex, it is more likely to be accurate than not; 3) user-defined priority of omics data: the user's confidence in the correctness of each omics type is documented as ranks. We manually identified 44 high-confidence mismatched data pairs with well-defined switch directions from the PsychENCODE BrainGVEX project[6] and used them to train the logistic regression model and establish parameters. (**Methods**). In the last step, we determine the final ID for all omics data points within each group. We sort all data points in each group using a modified topological sorting algorithm that is based on the switch probabilities obtained from the logistic regression above (**Methods**). The data ID with the highest value in each group will be selected as the final ID. In this study, we used simulation data and the PsychENCODE BrainGVEX data to evaluate the performance of DRAMS.

## Performance of DRAMS using simulation data

To test the performance of DRAMS on correcting data IDs, we generated multiple highly related data pair datasets with a few samples (ranging from 5% to 30%) being randomly shuffled to simulate sample mix up. The variable parameters for the simulation data include a range of sample sizes, numbers of omics types, and percentages of mix-ups (**Methods**). The stringent mode of DRAMS was used. In the stringent mode, we discarded the data groups with less than three data or with no shared IDs (i.e. all data IDs in a group are different), as they are almost unlikely to be corrected. We found that when a larger proportion of samples are mixed up, fewer samples can be successfully corrected (**Fig 2,** S3 Table). Taking datasets with five omics types and a sample size of 300 as examples, when 5% of the samples are mis-assigned, all samples can be successfully corrected; when 30% of the samples are mis-assigned we can successfully correct an average of 91.5% (SD: 2.30%) of errors, with an average of 0.481% (SD: 0.734%) of overcorrected items (**Fig 2C**).

The number of omics types involved also greatly influences the performance of DRAMS. The more omics types we have, the better the chance that we can recover the true identity of each erroneous data. Taking datasets with a sample size of 300 and 15% of mixed-up samples as examples, if we have three omics types (the minimum number of omics types required for DRAMS input), an average of 72.2% (SD: 4.62%) of mix-ups can be successfully corrected, with an average of 0.0207% (SD: 0.146%) of overcorrected items (**Fig 2A**). If six omics types are used, an average of 99.6% (SD: 0.815%) mix-ups can be successfully corrected, with an average of 0.119% (SD: 0.479%) of overcorrected iterms (**Fig 2D**).

Sample size is also important to the performance of DRAMS. Yet, sample size has no significant influence on DRAMS performance when proportions of sample mix-ups and number of

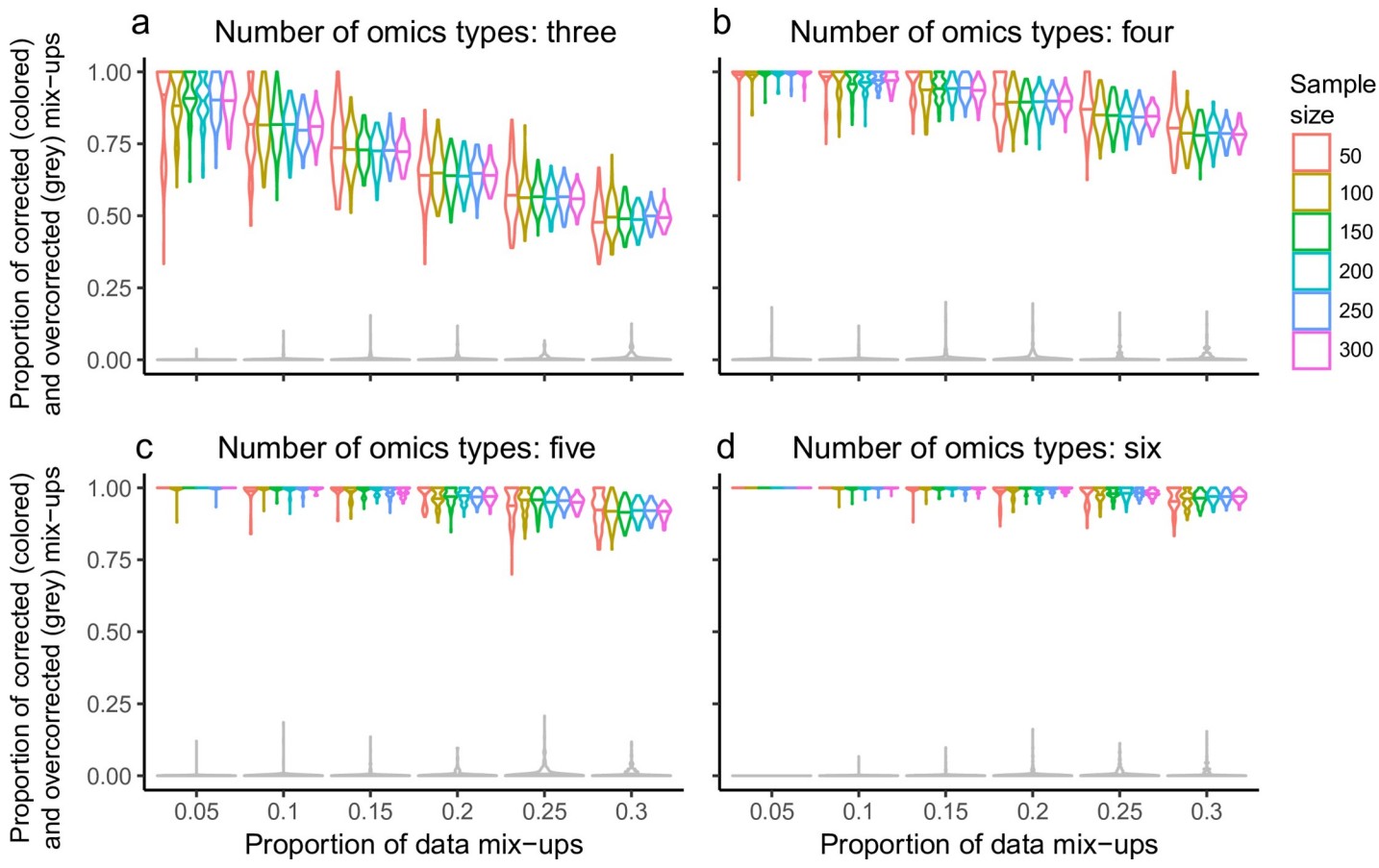

**Fig 2. Performance of DRAMS in simulation data.** We simulated sample mix-ups and used DRAMS to correct data IDs. The simulation data includes a range of sample sizes (50, 100, 150, 200, 250, and 300, each with 50% females and 50% males) and a range of omics types (3, 4, 5, and 6 for figure a, b, c, and d, respectively). To simulate sample mix-ups, we randomly shuffled a gradient proportion of data for each omics type: 5%, 10%, 15%, 20%, 25%, and 30%. The simulation process was repeated 100 times. For each repeat, we used DRAMS (the stringent mode) to correct mix-ups. The colored violin plots show the proportion of mix-ups successfully corrected (different colors indicate different sample sizes). The grey violin plots show the proportion of mix-ups overcorrected (simulations of different sample sizes were combined). The process of generating simulation data was described in S1 Fig.

omics used remain consistent (Pearson correlation between sample size and average proportions of successfully corrected mix-ups: -0.0105; P value: 0.901). However, larger sample sizes seem to stabilize correction results. As sample size increases, the standard deviation of the proportions of successfully corrected mix-ups decreased (Pearson correlation: -0.395; P value: $9.50 \times 10^{-7}$).

## Performance of DRAMS using real data from the PsychENCODE BrainGVEX project

**Data summary.** The PsychENCODE BrainGVEX project[6] generated six types of omics data (S1 Table), including low-depth Whole Genome Sequencing (WGS), RNA sequencing (RNA-Seq), Assay for Transposase-Accessible Chromatin using Sequencing (ATAC-Seq), Ribosome Sequencing (Ribo-Seq), and SNP array data from two platforms, including Affymetrix 5.0 450K (Affymetrix) and Psych v1.1 beadchips (PsychChip). We called a total of 19,242,755 SNPs from WGS data on autosomes, 17,786,350 SNPs from RNA-Seq data, 10,571,742 SNPs from ATAC-Seq data, 156,354 SNPs from Ribo-Seq data, 10,891,109 SNPs from Affymetrix data, and 13,589,867 SNPs from PsychChip data. We used two methods to

check sample contamination: using VerifyBamID[7], and calculating heterozygous rates (S2 Fig). We defined the samples with both FREEMIX >0.3 and heterozygous rates >0.3 as contaminated samples. We removed the sample "2015–916" in WGS from the subsequent analyses for being contaminated.

**Check sample alignment based on genetics-based sex.** We checked sample alignment by comparing genetics-based sexes with reported sexes for data of WGS, PsychChip, ATAC-Seq, and RNA-Seq. Based on X chromosome heterozygosity and Y chromosome call rate, we calculated an F-value (A Plink-derived metric to distinguish males and females. Details in Methods) for each data in each omics type. Then, the genetics-based sexes were inferred according to the distribution of F-values (called SNP-inferred sexes). By comparing reported sex with SNP-inferred sex, we identified a total of 74 data with mismatched sexes and 1174 data with matched sexes, indicating that some samples might have been mixed-up (**Table 1**). For the 426 samples in RNA-Seq, only three samples were identified as having mismatched sexes, indicating that this RNA-Seq data may have an overall good quality in terms of sample matching. For Ribo-Seq data, we did not estimate SNP-inferred sexes as the threshold to separate males and females, since Ribo-Seq data cannot be defined based on the distribution of F-values. Neither did we estimate SNP-inferred sexes for Affymetrix 5.0 SNP array data since no genotype on sex chromosomes were available. As complementary evidence, we also used *XIST* gene expression level (XIST-inferred sexes) to infer genetics-based sexes for RNA-Seq and Ribo-Seq data (**Methods**).

**Detect and correct data IDs.** We calculated genetic relatedness scores among all the data in the six omics types using GCTA[3]. Based on the distribution of genetic relatedness scores, we extracted the highly related data pairs using a threshold of 0.65 (S3 Fig). We identified a total of 1971 matched pairs and 518 mismatched pairs (**Fig 3**). We also found eight data that were not related to any other data, including three ATAC-Seq data, four Ribo-Seq data, and one Affymetrix data (S4 Table).

Of the 518 mismatched data pairs, 44 pairs have certain switch directions. We used these to train the logistic regression model (**Method**). For the remaining 474 mismatched data pairs, we used the logistic regression model to predict switch directions and probabilities. Because WGS and PsychChip data were processed with the same sources of DNA, we considered them as one omics type in the regression step. Similarly, we considered ATAC-Seq and Ribo-Seq as one omics type in the regression, as they were processed from the same original tissues. Based on the proportion of samples with concordant SNP-inferred sex and reported sex (**Table 1**), as well as on our prior knowledge about the data processing of each omics type, we assigned the

**Table 1. Summary of samples and sample corrections in BrainGVEX.**

| Data type | Number of samples | Number of contaminated samples | Number of sex-matched samples* | Number of sex-mismatched samples | Number of samples missing sex information | Proportion of sex-matched samples | Number of samples switched IDs | Number of samples unrelated to any other | Number of Sex-mismatched samples after correcting IDs |
|---|---|---|---|---|---|---|---|---|---|
| WGS | 285 | 1 | 256 | 24 | 5 | 0.914 | 54 (19.0%) | 0 | 0 |
| PsychChip | 263 | 0 | 244 | 19 | 0 | 0.928 | 43 (16.3%) | 0 | 0 |
| Affymetrix | 137 | 0 | - | - | - | - | 0 | 1 | - |
| ATAC-Seq | 295 | 0 | 260 | 28 | 7 | 0.903 | 55 (18.6%) | 3 | 15 |
| RNA-Seq | 426 | 0 | 414 | 3 | 9 | 0.993 | 3 (0.7%) | 0 | 3 |
| Ribo-Seq | 197 | 0 | - | - | - | - | 50 (25.4%) | 4 | - |
| Total | 1603 | 1 | 1174 | 74 | 21 | - | 201 (12.5%) | 8 | 18 |

*Note*: "Number of sex-matched samples" indicates the number of samples with the same reported sex and SNP-inferred sex.

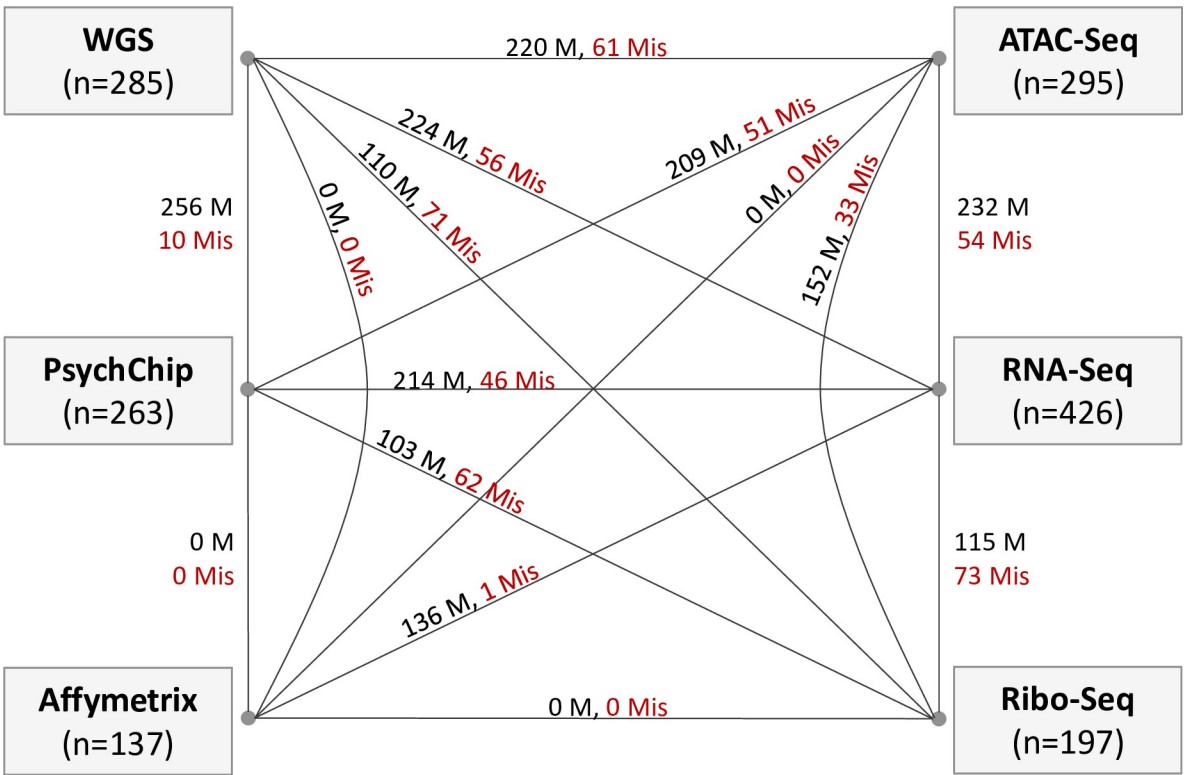

**Fig 3. Summary of highly related data pairs.** Summary of highly related data pairs among the six omics types. A highly related data pair was defined as a data pair with genetic relatedness score > 0.65. M: Matched pairs (highly related data pairs that have the same individual IDs), Mis: Mismatched pairs (highly related data pairs with different individual IDs).

omics priority as RNA-Seq > Affymetrix > WGS & PsychChip > ATAC-Seq & Ribo-Seq. After using the logistic regression to predict the switch directions and probabilities for the 474 mismatched data pairs, we connected all highly related data pairs to create groups of paired data. Then, we used the modified topological sorting algorithm to sort the nodes in each group and picked the node with the highest score as the final ID for all data in each group. In the end, we corrected 201 (12.5%) IDs for data of the six omics types (**Table 1,** S5 Table).

After correcting data IDs, eighteen data still have mismatched SNP-inferred sexes and reported sexes, including 15 ATAC-Seq data and three RNA-Seq data (S4 Fig). For ATAC-Seq, mismatches are not unexpected since accurately inferring genetics-based sex for some samples is difficult due to their low sequencing coverage in X and Y chromosomes. For RNA-Seq, we found that the XIST-inferred sexes were all consistent with reported sexes for all the samples (S5 Fig), indicating that all the samples in RNA-Seq might have been assigned with their true IDs. As we found three samples with mismatched SNP-inferred sexes and reported sexes, we inferred that it may be inaccurate to estimate genetics-based sexes based on the genotypes called from RNA-Seq data. For Ribo-Seq, both XIST-inferred sexes and F-values reported sexes were more inconsistent than RNA-Seq or DNA-based data. This may suggest that neither sex chromosome genotypes nor XIST gene expression works well to infer genetics-based sexes for Ribo-Seq data.

**Validate data ID corrections by race group assignment.** To confirm that the 201 data IDs were correctly assigned, we used race as an independent validation. We performed Principal Component Analysis (PCA) on data of the four major racial groups, European, Asian, African, and African American from the 1000 Genomes Project (1000G)[8] (S2 Table) and our

BrainGVEX data. PCA plotted the 1000G and BrainGVEX data into four racial groups. Before correcting data IDs, twenty-one data were classified in wrong racial groups. After correcting data IDs, all the data have concordant races with 1000G (S5 Table). For WGS, PsychChip, ATAC-Seq, and Ribo-Seq data that have race-switched data IDs, all were switched back into the correct PCA groups, indicating that those samples were likely to have been mislabeled and successfully corrected by DRAMS (**Fig 4**).

**Increased number of cis-QTLs after correcting data IDs.** We mapped four sets of cis-QTLs based on different data combinations (WGS with RNA-Seq, WGS with Ribo-Seq, PsychChip with RNA-Seq, and PsychChip with Ribo-Seq) for BrainGVEX data before and after correcting data IDs using DRAMS. After correcting data IDs, although the sample sizes were reduced slightly due to the removal of a few unresolved samples, the numbers of cis-QTLs increased by an average of 1.62-fold for the FDR<0.01 cutoff and average 1.54-fold for the FDR<0.05 (**Table 2**). We also tested the proportion of novel and discarded eQTLs replicated in the Genotype-Tissue Expression (GTEx) project [9] (**Table 2**). Around 50% of the novel eQTLs can be replicated in GTEx (denoted by π1). For the discarded eQTLs, only around 20% can be replicated. This relative stability clearly demonstrated the power and importance of correcting data IDs in QTL mapping.

## Sensitivity and specificity in extracting highly related data pairs

To determine the minimum number of SNPs needed to accurately identify highly related data pairs from all random pairs, we randomly selected eight subsets of SNPs numbered from 200 to 10,000 from the BrainGVEX data and re-calculated pair-wise genetic relatedness scores. Five types of omics data were used for this estimation, including WGS, PsychChip, ATAC-Seq, RNA-Seq, and Ribo-Seq. Only the samples with fully matched data IDs among all omics types were used.

Since both the WGS and the PsychChip platforms cover most of the common SNPs in the whole genome, we were able to successfully identify all the highly related data pairs without any false positives, using as few as 200 common SNPs (MAF > 0.1) (S6 Table, S6 Fig). When comparing DNA-based genotypes with data from other platforms, such as RNA-Seq, Ribo-Seq, or ATAC-Seq, we could capture only a small proportion of genotyped loci ranging from 74% to 85%, due to the platforms' limited coverage of genomic regions. To ensure that enough genotypes are compared, at least 1000 SNP loci should be used to estimate genetic relatedness.

The comparison of data from RNA-Seq, Ribo-Seq, or ATAC-Seq can be problematic since each platform has somewhat different priorities for capturing various genomic regions. Because of this, we found that fewer genotyped loci could be compared, undoubtedly reducing sensitivity and specificity. For example, for the comparison between RNA-Seq and ATAC-Seq data, the proportion of shared SNPs range from 51% to 55%. Even when 1000 common SNPs were used, nine false positive pairs and three false negative pairs were still found (sensitivity: 0.9796; specificity: 0.9996). We recommend using 2000 or more common SNPs to estimate genetic relatedness.

We also did the same analysis using rare SNVs (MAF < 0.1). Rare SNVs are less powerful to distinguish highly related data pairs from random pairs than common SNPs (S7 Table, S7 Fig). It is difficult to identify all highly related data pairs even when 10,000 rare SNVs were used. However, rare SNVs have good specificity. When 1000 rare SNVs were used, only two false positives for WGS versus ATAC-Seq (specificity: 0.9999), and five false positives for RNA-Seq versus ATAC-Seq (specificity: 0.9998) were found. No false positives were found for all the other comparisons.

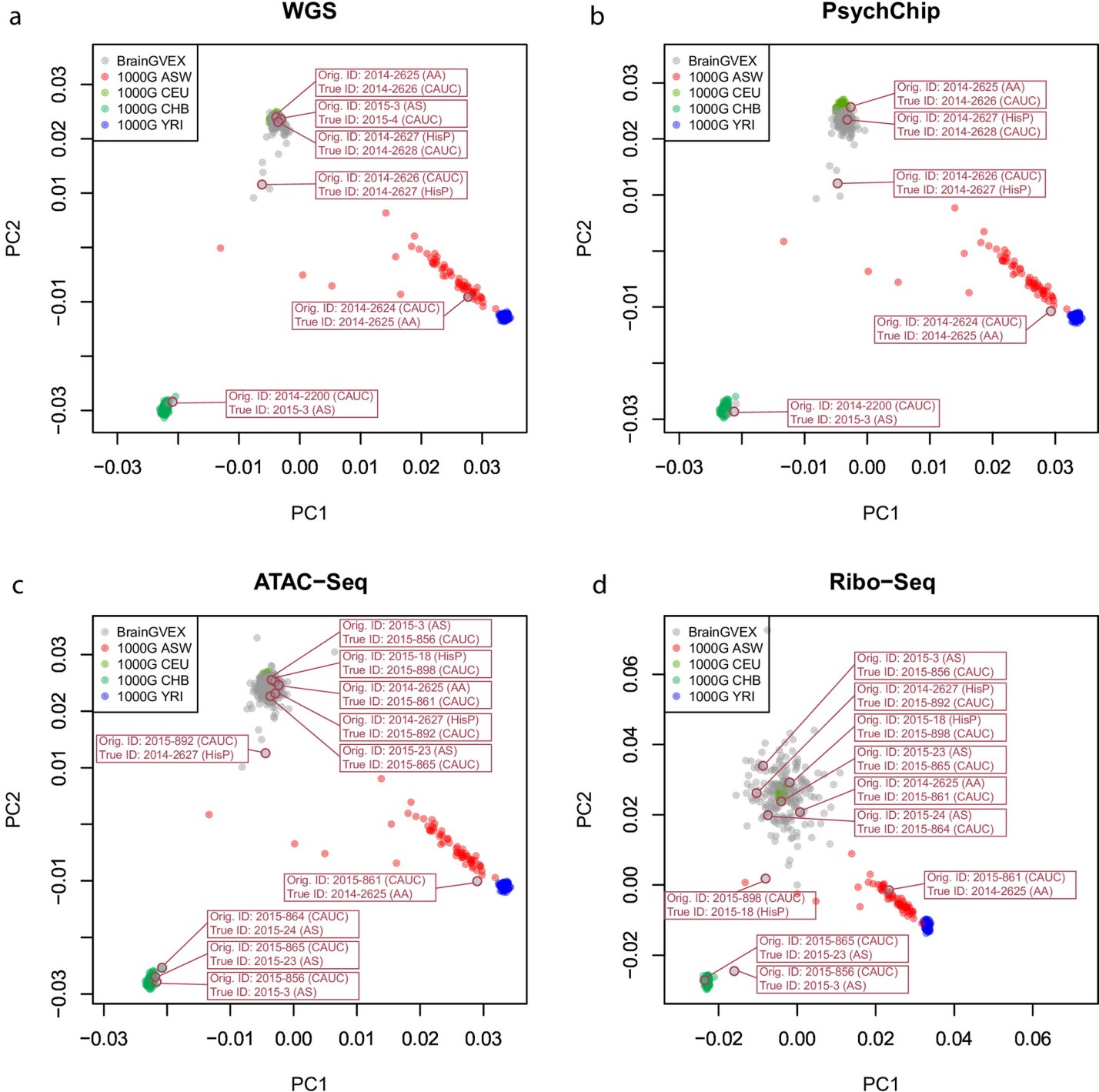

**Fig 4. Validation of cross-races switched data.** PCA results for BrainGVEX samples (grey dots) and 1000G samples (colored dots) are shown. All data with switched races are marked with their original IDs and new IDs, as well as the corresponding race. The correspondence between BrainGVEX and 1000G races are shown in S2 Table. PCA results for BrainGVEX (a) WGS, (b) PsychChip, (c) ATAC-Seq, and (d) Ribo-Seq samples are shown. RNA-Seq and Affymetrix data are not shown as there is no data with switched races. Proportions of switched races data per dataset: WGS: 0.12; PsychChip: 0.12; ATAC-Seq: 0.19; Ribo-Seq: 0.21. CAUC: Caucasian, HiSP: Spanish, AA: African American, AS: Asian American, CEU: Utah Residents with Northern and Western European Ancestry, ASW: African Ancestry in Southwest US, CHB: Han Chinese in Beijing, China, YRI: Yoruba in Ibadan, Nigeria.

**Table 2. Increased number of cis-QTLs after correcting data IDs.**

| QTL type | Category | Before correcting IDs | After correcting IDs | Fold change | Novel QTLs after correcting IDs ($\pi_1$ in GTEx [*]) | Discarded QTLs after correcting IDs ($\pi_1$ in GTEx) |
|---|---|---|---|---|---|---|
| WGS vs. RNA-Seq | Sample size | 278 | 273 | - | - | - |
| | #QTLs (FDR<0.01) | 57,209 | 96,242 | 1.68 | 43,266 (0.608) | 4,233 (0.246) |
| | #QTLs (FDR<0.05) | 90,231 | 147,942 | 1.64 | 66,993 (0.475) | 9,282 (0.213) |
| WGS vs. Ribo-Seq | Sample size | 191 | 187 | - | - | - |
| | #QTLs (FDR<0.01) | 18,178 | 31,345 | 1.72 | - | - |
| | #QTLs (FDR<0.05) | 30,641 | 48,306 | 1.58 | - | - |
| PsychChip vs. RNA-Seq | Sample size | 259 | 253 | - | - | - |
| | #QTLs (FDR<0.01) | 48,742 | 76,995 | 1.58 | 31,801 (0.638) | 3,548 (0.581) |
| | #QTLs (FDR<0.05) | 77,925 | 117,711 | 1.51 | 49,682 (0.519) | 9,896 (0.246) |
| PsychChip vs. Ribo-Seq | Sample size | 177 | 172 | - | - | - |
| | #QTLs (FDR<0.01) | 15,028 | 20,447 | 1.36 | - | - |
| | #QTLs (FDR<0.05) | 26,350 | 32,209 | 1.22 | - | - |
| Total | #QTLs (FDR<0.01) | 139,157 | 225,029 | 1.62 | 59,399 | 17,020 |
| | #QTLs (FDR<0.05) | 225,147 | 346,168 | 1.54 | 95,495 | 34,684 |

*Note*: Only chromosome 1 was used to save computing time.

[*] $\pi_1$ was estimated using the "qvalue" package in R.

## Discussion

To meet the demand for reducing the influence of sample mix-ups on multi-omics integrative studies, we developed a tool DRAMS to detect and correct mixed-up data IDs. The principle of DRAMS is that genotypes of all omics data assayed on the same individuals should be identical. We directly call genotypes and estimate pair-wise genetic relatedness by calculating the genotype concordance rates among all data to be checked. Therefore, any omics type, as long as it contains genotype information, could be used for DRAMS correction.

DRAMS groups the data potentially originating from the same individual together and determines the potential IDs for data within each group. The group size is influenced by the number of omics types. Having more omics types bolstered the information available to unlock the potential IDs. However, the increase also results in greater complexity. DRAMS used a logistic regression model followed by a modified topological network sorting algorithm to systematically integrate the genetic relationships, sex concordance, and omics priority to determine the potential ID for each data. The tool performs well in both simulation data and BrainGVEX data. With this design, DRAMS can be applied to an unlimited number of omics data. According to our simulation data, DRAMS performs better as more omics types are included. This is a major advancement of our framework that outperformed existing tools.

Since sex plays such an essential role in verifying data ID, we naturally chose to employ sex information in the design of DRAMS to increase reliability. When applying DRAMS to

 

BrainGVEX data, we used two strategies; we inferred genetics-based sexes from sex chromosome genotypes (SNP-inferred sex) and alternatively, from XIST gene expression (XIST-inferred sex). When applying DRAMS to RNA-Seq data, all data had consistent XIST-inferred sexes and reported sexes after correcting the data IDs. However, three out of 426 data had inconsistent SNP-inferred sexes and reported sexes. For Ribo-Seq data, as XIST is a non-coding RNA, it's not accurate to estimate genetics-based sex according to XIST expression. Also, due to the low coverage for Ribo-Seq data, estimating genetics-based sex according to sex chromosome genotypes is not confident. Therefore, it is reasonable that we did not see an obvious consistency among XIST-inferred sexes, SNP-inferred sexes, and reported sexes.

We used ethnic information as a validation step for BrainGVEX data ID correction. After correcting data IDs, all of the samples grouped into the correct race with other 1000G reference samples. Although matched race does not necessarily mean correct data ID, it is strong evidence to prove that the data IDs were switched to the correct directions for the mismatched data pairs.

We identified more QTLs after data ID correction. Although this is not direct proof for correct data ID assignment either, it is likely the results of better alignment of different omics types, and consequently largely increased statistical power for QTL analyses.

The threshold to extract highly related data pairs should be selected very carefully. A loose threshold can lead to a large proportion of overcorrection (S8 Table). For data of related individuals such as family data, it is possible that the related individual pairs have the relatedness score > 0.65. For this situation, we recommend to use a stringent threshold to extract highly related data pairs using GCTA or NGSCheckMate. For GCTA, we recommend to eyeball the distribution of genetic relatedness scores and choose a higher threshold. For NGSCheckMate, the tool provided a "-f" parameter that defines highly related sample pairs in a stringent mode, which will reduce the probability of mislabeling among relatives. However, even in stringent mode, it is still possible that some related individuals be identified as the same individual. In other words, DRAMS need to be used with great caution when processing data from family members. That will be a major challenge for algorithm, particularly on data with less usable genotypes or of more noise.

We used the BrainGVEX data to assess the minimum number of SNPs needed to identify highly related data pairs. We found that common SNPs have greater power in distinguishing highly related data pairs from random pairs. Nonetheless, rare SNVs are also useful as they are unlikely to produce false positive findings. In addition, we also found that when comparing data from different platforms (which cover different genomic regions), a smaller proportion of genotyped loci could be compared between different platforms, indicating that additional SNPs should be used. Based on the BrainGVEX data, we recommend using 2000 or more common SNPs to extract highly related data pairs for most platforms. Nonetheless, since the genotype qualities of different platforms may differ substantially, we recommend including as many variants as possible, even rare variants, to fortify the sensitivity and specificity of the DRAMS tool.

Matching omics data is only the first step in the process. Assigning the correct data ID, typically associated with sample demographic information and phenotypic data, is another important step. Conceivably, some analyses are more sensitive to sample information, covariates (sex, diagnosis, etc.) than others. Mis-assigned sample information severely affects some analyses, such as differential gene expression and case-control comparison. DRAMS can correct mix-ups and identify correct labels and associated sample information, before conducting the integrative analyses.

Currently, only a few projects have produced multi-omics data, like the data generated in Drs. Gilad and Pritchard's lab[10], data from GTEx[9], and ROSMAP[11, 12]. As more large-

 

scale multi-omics data will be generated in the near future to address problems of regulatory networks and causal relationships, we expect that DRAMS will be helpful for those studies.

## Materials and methods

### Sample resources

A total of 440 individuals from the PsychENCODE BrainGVEX study[6] with six types of omics data were used to validate data IDs and assign the potential IDs. BrainGVEX was part of the PsychENCODE project focusing on gene expression regulation in human brain FC region (Frontal Cortex). The samples include 420 Caucasians, 2 Hispanic, 1 African American, 3 Asian American, and 14 unknown (S1 Table). The six omics types included 1) 285 samples (176 males, 106 females, and 3 unknown-sex samples) of low-depth Whole Genome Sequencing (WGS, average depth: 5×) data; 2) 426 samples (274 males, 152 females) of RNA-Seq data; 3) 295 samples (180 males, 112 females, and 3 unknown-sex samples) of Assay for Transposase-Accessible Chromatin using Sequencing (ATAC-Seq) data,; 4) 197 samples (122 males, 70 females, and 5 unknown-sex samples) of Ribosome Sequencing (Ribo-Seq) data; and, SNP array data from two platforms, including 5) 137 samples (92 males, 45 females) of Affymetrix 5.0 450K (Affymetrix) data; and, 6) 263 samples (163 males, 100 females) of Psych v1.1 beadchips (PsychChip) data.

### Genotype calling from data of each omics type

We used the same pipeline to call genotypes for all sequencing data. For each dataset in FASTQ format, all reads were mapped to the human reference genome (hg19) using BWA [13] after sequencing adapters and low-quality bases were removed. PCR duplications were removed using the MarkDuplicates package in Picard tools (http://broadinstitute.github.io/picard/). Then, GATK IndelRealigner and BaseRecalibrator were used to recalibrate the mapping quality of the reads [14]. For each omics type, genotypes were called using GATK HaplotypeCaller for all samples jointly. Each set of omics data were processed separately.

### Estimation of sample contamination

Two methods were used to check sample contamination. One is VerifyBamID[7], which because it requires both BAM files and VCF files as input, can only be applied to sequencing data. The results include a parameter "FREEMIX" (0–1 scale), which indicates the proportion of non-reference bases observed in reference sites. This parameter can be used as an indicator of sample contamination. As an alternate method, we wrote a Linux script to directly calculate the heterozygous rate based on genotypes. This ran faster than the VerifyBamID approach. We defined the samples with FREEMIX >0.3 in VerifyBamID and defined heterozygous rates >0.3 as contaminated samples, which were removed from subsequent analyses. Only heterozygous rates were calculated for PsychChip and Affymetrix samples, as they are not supported by VerifyBamID.

### Infer genetics-based sexes

We used the Plink software ("—check-sex" module in "ycount" mode) to calculate F-values for data of WGS, PsychChip, ATAC-Seq, RNA-Seq, and Ribo-Seq [15]. This method is mainly based on the X chromosome heterozygosity. It also uses the Y chromosome call rate to improve the accuracy of sex estimates. Basically, the F-values were in bi-modal distribution. Based on the distribution, we were able to select a threshold for each omics type and infer sexes (SNP-inferred sexes) for each data. The data with F-value larger than the threshold were

classified as males, while the others were classified as females. We did not infer SNP-inferred sexes for Ribo-Seq data since an obvious bi-modal distribution of F-values was not apparent.

For RNA-Seq and Ribo-Seq data, we also inferred sexes based on XIST (X-inactive specific transcript) gene expression levels (XIST-inferred sex). XIST is a noncoding RNA that is only expressed in cells containing at least two X chromosomes [16]. Normally, the XIST gene is only expressed in female samples. In this study, we considered samples with XIST expression larger than 2 (TPM, Transcripts Per Kilobase Million) as females.

We compared the reported sex and genetics-based sex for each data in each omics type and calculated the sex concordance rate for each omics type. Samples with unknown SNP-inferred sexes were not included when calculating the sex concordance rate. The sex concordance rate can represent a parameter indicating omics priority in the logistic regression model (See "Estimate switch directions and probabilities for mismatched data pairs").

## Estimate genetic relatedness and extract highly-related data pairs

Two tools were used to estimate genetic relatedness among data of multiple omics types and extract highly-related data pairs: GCTA[3] and NGSCheckMate[4]. For GCTA, GRM module was used. Basically, the genetic relatedness scores are distributed bimodally, so that one peak with higher scores indicates highly-related data pairs while the other peak indicates random (unrelated) data pairs. In this way, one can "eyeball" the distribution to determine the thresholds of genetic relatedness scores between every two omics datasets (**Fig 1**). Using that method, we applied a threshold of 0.65 for our BrainGVEX data and extracted highly related data pairs in different omics types. For NGSCheckMate, we ran the software in VCF mode with an "-f" parameter to enact a strict VAF correlation filter. We calculated the concordance rates of the two tools based on BrainGVEX data.

## Sensitivity and specificity in extracting highly related data pairs

To determine the minimum number of variants required to extract highly related data pairs from all combination of data pairs, we re-calculated genetic relatedness scores using the BrainGVEX data based on subsampled SNPs. Data from five omics types were used, including WGS, PsychChip, ATAC-Seq, RNA-Seq, and Ribo-Seq. We used only the samples that fully matched in all the five omics types. For comparisons between each two omics types, we randomly selected 200, 400, 600, 800, 1000, 2000, 5000, and 10,000 SNPs that were called by both omics types and calculated pair-wise genetic relatedness scores using GCTA. The data pairs with genetic relatedness scores larger than 0.65 were classified as *highly related data pairs* and those pairs with smaller scores were classified as *unrelated data pairs*. For each comparison, we calculated the true positive rate and false negative rate. We did the same analysis for common (MAF>0.1) and for rare (MAF<0.1) variants.

## Estimate switch directions and probabilities for mismatched data pairs

To estimate the possible switch directions and the probabilities for a mismatched data pair, the key point must be focusing on determining which data are more likely to bear the true ID. We used a logistic regression model (Formula 1) to compare the two data presented in each mismatched data pair and to estimate the possible switch direction and its probability (**Fig 1**). Assuming the mismatched data pair "A" and "B", if "B" is more likely to have the true ID, then the appropriate switch direction is from "A" to "B". Three parameters ($x_1$, $x_2$, and $x_3$, values range from 0 to 1) were used in this model. The first parameter, $x_1$, indicates which data matched with more data in other omics types (Formula 2). The second parameter, $x_2$, indicates which data are more likely to have correct reported sex (Formula 3); and the third, $x_3$, is a

user-defined parameter indicating the rank of omics priority. If two or more omics types were processed under the same condition or process, these omics types may have the same sample mis-labeling, a user could combine these omics types into one type in the regression. If $x_3$ is not specified, the rank of omics priority will be defined based on the proportion of data that have matched reported sex and genetics-based sex for each omics type (Formula 4).

$$ln\left(\frac{p}{1-p}\right) = \beta_0 + \beta_1 x_1 + \beta_2 x_2 + \beta_3 x_3 \tag{1}$$

$$x_1 = \frac{n_b - n_a}{N - 2} \tag{2}$$

$$x_2 = S_b - S_a \tag{3}$$

$$x_3 = P_b - P_a \tag{4}$$

In formula 2, N is the total number of omics types; and correspondingly, $n_a$ and $n_b$ are the number of matched data from other omics types for data "A" and "B", respectively. In formula 3, $S_a$ and $S_b$ indicate the sex matching level for data "A" and "B", respectively. Taking $S_a$ as an example, it indicates 1) whether the reported sex and genetics-based sex in data "A" are matched, and 2) if assigning ID "A" to data "B", do the reported sex of data "A" and the genetics-based sex of data "B" match? We assign a score 0.5 for each of the two conditions. In formula 4, $P_a$ and $P_b$ represent the proportion of data with matched reported sex and genetics-based sex for data "A" and "B", respectively.

A set of hand-picked high-confidence mismatched data pairs with well-defined switch directions were used as a training set for the logistic regression model. The directions of the high-confidence mismatched data pairs were determined based on sample relationships and sex matching. We connected all the highly related data pairs among multiple omics types and created multiple groups. We extracted the high-confidence mismatched data pairs based on the following conditions: 1) If only one data had a different ID, which needs to be corrected, from the others in the group; 2) If the reported sex and genetics-based sex were matched after correcting that ID based on other data in the group.

After training, the values for $\beta_0$, $\beta_1$, $\beta_2$, and $\beta_3$ were defined. Then, the model was used to predict switch directions and probabilities for the rest of the data. For the results, if $p > 0.5$, the switch direction will be from "A" to "B"; conversely, if $p < 0.5$, the switch direction will be from "B" to "A"; and, if $p = 0.5$, the switch direction will be uncertain. The values of p for both directions ($p > 0.5$, $p < 0.5$) were normalized (with a range from 0 to 1 respectively) to indicate the probabilities of switch directions.

## A modified topological sorting algorithm to determine the potential IDs

We connected all highly related data pairs and generated multiple groups. In each group, each data was represented by a node, and each data pair was represented by an edge. Based on the logistic regression model results, the switch direction and probability for each mismatched data pair corresponded to the direction and weight of each edge. For each matched data pair, we did not assign a direction or weight for the edge. Since all the data presented in a group were supposed to have one unique ID, we used a modified topological sorting algorithm to sort all nodes in each group to determine the potential ID for each data.

The modified topological sorting algorithm was based on the indegrees and outdegrees weighted by the switch probabilities which had been calculated in the logistic regression

model. For each node in each group, we calculated the difference between weighted indegree and weighted outdegree. Afterward, we sorted the nodes in each group based on the difference between weighted indegrees and outdegrees. For each group, we used the data ID with the highest priority as the final ID for all data presented in the group.

## Simulation data

We generated simulation data to test the performance of DRAMS on correcting data IDs (S1 Fig). At first, we generate data IDs for different samples and omics. A range of sample sizes (50, 100, 150, 200, 250, and 300) was used, with each dataset having 50% females and 50% males. The number of omics types ranged from three to six. Then, we randomly shuffled parts (5%, 10%, 15%, 20%, 25%, and 30%) of the data IDs. In total, we generated 144 simulated datasets (six sets of sample size × four sets of omics type × six proportions of mixed-up sets). In an attempt to mimic reality, we randomly introduced mislabeled sexes for 2% of samples in our simulation data. In addition, since samples are often swapped within the same batch in reality, we divided the samples into several batches, with each batch containing 25 samples. The data ID shuffling all occurred within batches. For each set of simulation data, we corrected data IDs using DRAMS (stringent mode) and calculated the proportion of mix-ups being successfully corrected or overcorrected. In stringent mode, we discarded the data groups with less than three data or with no shared IDs (all data IDs in a group are different). We simulated the process 100 times for each dataset.

## Validate data ID corrections by racial assignment

To validate the tool, we started by using genotypes that shared loci in BrainGVEX and the 1000 Genomes project (1000G)[8]. For 1000G, we used only the following samples: ASW (African Ancestry in Southwest US), CEU (Utah Residents with Northern and Western European Ancestry), CHB (Han Chinese in Beijing, China), and YRI (Yoruba in Ibadan, Nigeria). We first removed genotypes with MAF < 0.01. Then, we used GCTA[3] to estimate genetic relationships among individuals and performed PCA analysis for both BrainGVEX and 1000G samples jointly. We used data from 1000G that were located in the same group as the reference, after which we compared the races of samples from BrainGVEX with the reference before and after correcting data IDs. Since BrainGVEX and 1000G used different nomenclatures for races, we used analogical names or close races between BrainGVEX and 1000G to align the data (S2 Table).

## Detect QTLs before and after correcting mix-ups

We used FastQTL[17] to map QTL within the BrainGVEX samples both before and after correcting data IDs. We defined the cis-QTL region as 1 million base pairs between the SNP marker and the gene body. Since we intend to test whether the number of cis-QTLs increased, only chromosome 1 was used to save computing time. R package "qvalue"[18] was used for multiple tests. We tested four types of cis-QTLs calculations: WGS with RNA-Seq, WGS with Ribo-Seq, PsychChip with RNA-Seq, and PsychChip with Ribo-Seq. For RNA-Seq and Ribo-Seq samples, we used log2 transformed CPM quantification data calculated by VOOM[19]. We selected 30 hidden factors as covariates using the PEER software[20] for RNA-Seq and Ribo-Seq samples. As one or multiple of the hidden factors estimated by PEER were significantly associated with known covariates (age of death, diagnosis, brain bank, ethnicity, and sex) (S8 Fig), we only included the 30 PEER factors in QTL analyses. We used two cutoffs (FDR < 0.05, FDR < 0.01) for the cis-QTL results.

## Code availability

The code of DRAMS was implemented in Python3 and deposited in GitHub (https://github. com/Yi-Jiang/DRAMS). We released the data preprocessing codes, including genotype calling, sample contamination checking, genetics-based sex inference, genetic relatedness score calculation, and extracting highly related data pairs. We also provided a guideline for using Cytoscape to visualize sample relationships within networks [21].

## Computational requirement and timing for 200 samples with 6 omics data

Step 1, map reads to reference genome (FASTQ to BAM, for one sample, 4Gb data size): 4 CPU, 15G memory, 1.5 hours.

Step 2, call genotypes (BAM to VCF, for one sample, 4Gb data size): 12 CPU, 20G memory, 7 hours.

Step 3, infer genetics-based sexes (for one sample): 1 CPU, 1G memory, 1 minute.

Step 4, calculate genetic relatedness scores: 1 CPU, 1G memory, 10 minutes.

Step 5, correct data IDs: 1 CPU, 1G memory, 5 minutes.

## Supporting information

**S1 Fig. A flowchart of testing DRAMS in simulation data.** Simulation data were generated to test the performance of DRAMS. Step 2 to step 4 were repeated four times.
(TIF)

**S2 Fig. Heterozygous proportion and VerifyBamID results implied possible contaminated samples.** Two methods (Heterozygous proportion and VerifyBamID) were used to estimate sample contamination for WGS, ATAC-Seq, RNA-Seq, and Ribo-Seq data. For VerifyBamID results, "FREEMIX" (0–1 scale) was used to indicate possible sample contamination. For PsychChip and Affymetrix samples, we calculated only heterozygous proportions.
(TIF)

**S3 Fig. Distribution of genetic relatedness scores among omics types.** Genetic relatedness scores were calculated by GCTA.
(TIF)

**S4 Fig. Increased concordance of reported sex and genetics-based sex in corrected data IDs.** Genetics-based sexes were inferred using Plink. Larger F-value indicated that the sample is more likely to be male. Ribo-Seq and Affymetrix samples were not shown since we were not able to infer the genetics-based sexes.
(TIF)

**S5 Fig. Comparison of genetics-based sexes inferred from sex chromosome genotypes and *XIST* expression.** For sex chromosomes-inferred sexes, larger F-value indicates that the sample is more likely to be male. For *XIST* expression-inferred sex, the samples with *XIST* expression larger than zero are more likely to be female. The reported sexes were based on samples before ID correction.
(TIF)

**S6 Fig. Genetic relatedness scores calculated using subsampled common SNPs.** Genetic relatedness scores were calculated based on common SNPs (MAF>0.1) randomly selected

from BrainGVEX data. Only the samples that matched well among all omics types were used. We only showed a subset of randomly selected mismatched data pairs according to the number of matched data pairs.
(TIF)

**S7 Fig. Sample relatedness scores calculated using subsampled rare SNVs.** Genetic related-ness scores were calculated based on randomly selected rare SNVs (MAF<0.1) from BrainG-VEX data. Only the samples that matched well among all omics types were used. We only showed a subset of randomly selected mismatched data pairs according to the number of matched data pairs.
(TIF)

**S8 Fig. Correlation between known covariates and PEER factors.** Spearman correlation tests were performed between PEER factors and ageDeath. One-way ANOVA tests were performed between PEER factors and Diagnosis, BrainBank, Ethnicity, and Sex. P values were marked for the cells with significant correlation (P value < 0.05).
(TIF)

**S1 Table. Sample list in BrainGVEX.** A total of 440 samples from BrainGVEX were used to correct data IDs using our DRAMS. The number of data produced from different omics types are shown for each sample.
(XLSX)

**S2 Table. Correspondence between BrainGVEX and 1000G races.**
(XLSX)

**S3 Table. Proportion of successfully corrected mix-ups in simulation data.**
(XLSX)

**S4 Table. Highly related data pairs in BrainGVEX.** Genetic relatedness scores were calcu-lated using GCTA. The highly related data pairs were extracted using a 0.65 cutoff.
(XLSX)

**S5 Table. Corrected sample IDs in BrainGVEX.** We corrected a total of 201 sample IDs using DRAMS. PCA was performed for both BrainGVEX samples and 1000G samples using GCTA based on genetic relationships among the samples. The races in BrainGVEX samples and corresponding groups of 1000G were compared. The PCA results were shown in Fig 3.
(XLSX)

**S6 Table. Sensitivity and specificity in extracting highly related data pairs with subsampled common variants.** Only variants with MAF>0.1 were used. TP: true positive pairs, indicating data pairs with genetic relatedness score > 0.65 and with the same ID. TN: true negative pairs, indicating data pairs with genetic relatedness score < 0.65 and with different IDs. FP: false pos-itive pairs, indicating data pairs with genetic relatedness score > 0.65 and with different IDs. FN: false negative pairs, indicating data pairs with genetic relatedness score < 0.65 and with the same ID.
(XLSX)

**S7 Table. Sensitivity and specificity in extracting highly related data pairs with subsampled rare variants.** Only variants with MAF<0.1 were used.
(XLSX)

**S8 Table. The effect of different genetic relatedness thresholds on the performance of DRAMS.** The PsychENCODE BrainGVEX data were used to assess the effect of different

genetic relatedness thresholds on the performance of DRAMS. Only the samples with reported sex and race information and genetics-based sexes were used (WGS: 280, PsychChip: 263, RNA-Seq: 417, Ribo-Seq: 152, ATAC-Seq: 288). Gradient thresholds were used to extract highly related data pairs.
(XLSX)

**S9 Table. Comparison of GCTA and NGSCheckMate in estimating genetic relatedness.**
(XLSX)

**S1 Text. Comparison of GCTA and NGSCheckMate in estimating genetic relatedness.**
(DOCX)

## Acknowledgments

Thanks Liz Kuney and Richard Kopp for revising the manuscript. Thanks Hai Yang and Ying Ji for giving advice on presenting the work.

## Author Contributions

**Conceptualization:** Chao Chen, Chunyu Liu.

**Data curation:** Yi Jiang, Chunyu Liu.

**Formal analysis:** Yi Jiang.

**Funding acquisition:** Chao Chen, Chunyu Liu.

**Investigation:** Yi Jiang.

**Methodology:** Yi Jiang, Yan Xia, Quan Wang, Qiang Wei, Rui Chen, Bingshan Li, Chunyu Liu.

**Project administration:** Kevin P. White, Chunyu Liu.

**Resources:** Kevin P. White, Chunyu Liu.

**Software:** Yi Jiang, Lide Han, Sihan Liu.

**Supervision:** Chao Chen, Bingshan Li, Chunyu Liu.

**Validation:** Yi Jiang, Gina Giase, Kay Grennan, Annie W. Shieh.

**Visualization:** Yi Jiang, Lide Han.

**Writing – original draft:** Yi Jiang.

**Writing – review & editing:** Yi Jiang, Yan Xia, Chao Chen, Bingshan Li, Chunyu Liu.

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
