## [Decision Letter · Decision Letter 0]

27 Nov 2019

Dear Dr Liu,

Thank you very much for submitting your manuscript 'DRAMS: A Tool to Detect and Re-Align Mixed-up Samples for Integrative Studies of Multi-omics Data' for review by PLOS Computational Biology. Your manuscript has been fully evaluated by the PLOS Computational Biology editorial team and in this case also by independent peer reviewers. The reviewers appreciated the attention to an important problem, but raised some substantial concerns about the manuscript as it currently stands. While your manuscript cannot be accepted in its present form, we are willing to consider a revised version in which the issues raised by the reviewers have been adequately addressed. We cannot, of course, promise publication at that time.

Sincerely,

Ilya Ioshikhes

Associate Editor

PLOS Computational Biology

Thomas Lengauer

Methods Editor

PLOS Computational Biology

[LINK]

Reviewer's Responses to Questions

**Comments to the Authors:**

Reviewer #1: Jiang et al. developed a tool to detect and correct mixed-up samples in multi-omics data. The mix-up refers to switched IDs during the data generation process. Through analyzing the PsychENCODE BrainGVEX data, they corrected 12.5% mixed-up IDs, and this correction is shown to improve the discovery of cis-QTL. I think the paper addressed an important and interesting research question, and it was written in a convincing way. Below I list some comments that may improve the manuscript.

Major comments:

“We can cluster all the highly related data and consider that the data from one cluster have only one potential ID.”

It was not sure how the clustering was done. Is it based on a clustering algorithm? What is the resulted cluster size? Is it restricted to be the number of data types? Is it necessary that each cluster has each data type? More details are expected to explain the clustering procedure.

In Fig 1 and S2, the range of genetic relatedness score exceeds 1. Is it really possible to exceed one or just because of density curve smoothing? If the latter, the authors may consider using a way to estimate the density within a restricted range.

Additionally, in Fig 1, “The text in each node represents the data ID. Different colors represent different omics types.” The pair of A-A in Step 1 Type 1 are in the same color. Should A-A be in different colors?

In Fig 2, the authors assessed the proportions of successfully corrected mix-ups. I am wondering if there are non-mix-ups are wrongly classified as mix-ups and then overcorrected. For example, if using the cutoff of 0.65 of genetic relatedness scores for highly related data pairs, are there any non-mix-ups wrongly classified as mismatched data pairs? It may be related to the section of "Sensitivity and specificity in extracting highly related data pairs," but they are in different simulation settings.

In the simulation, how are the different data types simulated? How is the logistic regression fitted? Is it the same fitted model based on 44 data pairs as in the real data analysis?

The overcorrection may be an issue in the analysis of PsychENCODE data. “We identified a total of 1971 matched pairs and 518 mismatched pairs… In the end, we corrected 201 (12.5%) IDs for data of the six omics types.” There percentage of mismatched pairs and corrected IDs seems high.

Minor comments:

Fig 4 has a lot of abbreviations. Providing the full names in the caption would help reading.

The Supplementary Figures need captions.

“Since we intend to test whether the number of cis-QTLs increased, only chromosome 1 was used to save computing time.” This should be at least noted in the caption of Table 2 to help interpret the number of eQTLs.

Page 19, “In [Disp-formula pcbi.1007522.e003], Sa and Sb indicate the sex matching level for data “A” and “B”, respectively. Taking Sa as an example, **the original value of Sa would be 0**. If the reported sex and genetics-based sex in data “A” are matched, Sa would be **plus 0.5**. If the reported sex of data “A” and the genetics-based sex of data “B” are matched, which means that the reported sex and genetics-based sex are matched after switch ID from “B” to “A”, then Sa would be **plus 0.5**.” Could you explain more why the values of Sa are set in this way? It seems hard to understand and confusing.

Grammar error. Page 2: “As the number of datasets to be integrated increaseS”

Page 3: “all omics data of the same samples were clusterED together”

Page 4: “For the first step is to build highly related data pairs.”

Page 18: “those pairS with smaller scores were classified”

Reviewer #2: Comments to authors

The manuscript by Yi and colleagues, “DRAMS: A Tool to Detect and Re-Align Mixed-up Samples for Integrative Studies of Multi-omics Data”, presents a new method to detect and re-align mixed-up samples in multi-omics studies. The authors calibrated DRAMS using simulations and applied their method to the data from PsychENCODE BrainGVEX project to correct 201 sample IDs. They further tried to validated the results by comparing PCA and eQTL results before and after correcting sample IDs. The experiment is well designed and the analyses were carefully conducted. As the increasing emergency of multi-omics data, this method would be useful.

Major comments:

1. The authors should compare DRAMS with state-of-the-art methods of this kind in both simulation and real data analyses, such as MixupMapper.

2. DRAMS needs a set of hand-picked high-confidence mismatched data pairs as a training set. How many pairs with certain switch directions are required in the training process? What if there are no such mismatched data pairs?

3. The authors used a relatedness score of 0.65 to define highly related data pairs. Is it possible that the related individual pairs (due to relatedness rather than the same individual in different data types) have the relatedness score > 0.65? More justification is required here.

4. The authors claimed that more types of omics data were included, the power is higher. However, it would increase the computational complexity. In this case, how about the computational burden? It would be useful to summary the resources requirement (e.g., running time, memory usage) under different simulation scenarios.

5. In the eQTL analysis, did the authors correct for ancestry and sex? In addition, as discussed by the authors, large number of QTLs does not necessarily mean correct data ID. It would be useful to replicate the new discovery of eQTLs in an independent data set.

Minor comments:

1. Figure 1 is quite busy. It is difficult for the readers to follow. Detailed legend is needed.

2. Is it a typo of “404” mismatched data pairs, given 518 mismatched pairs in total and 44 pairs with certain switch directions?

3. In Table S4, there were 80 data pairs with the same ID were defined as “mismatched pairs”. It is not clear of the definition of “matched pairs” and “mismatched pairs” in Fig. 3. The authors also mentioned that eight data were not related to any other data. It does not make sense that only eight data are not related to any other data among all samples.

4. In the validation analysis using PCA, after correcting for mismatched IDS, the ancestry of all the other samples were matched correctly?

**Have all data underlying the figures and results presented in the manuscript been provided?**

Reviewer #1: None

Reviewer #2: Yes

PLOS authors have the option to publish the peer review history of their article (what does this mean?). If published, this will include your full peer review and any attached files.

Reviewer #1: No

Reviewer #2: No

---

## [Decision Letter · Decision Letter 1]

9 Feb 2020

Dear Liu,

Thank you very much for submitting your manuscript "DRAMS: A Tool to Detect and Re-Align Mixed-up Samples for Integrative Studies of Multi-omics Data" for consideration at PLOS Computational Biology. As with all papers reviewed by the journal, your manuscript was reviewed by members of the editorial board and by several independent reviewers. The reviewers appreciated the attention to an important topic. Based on the reviews, we are likely to accept this manuscript for publication, providing that you modify the manuscript according to the review recommendations.

Sincerely,

Ilya Ioshikhes

Associate Editor

PLOS Computational Biology

Thomas Lengauer

Methods Editor

PLOS Computational Biology

[LINK]

Reviewer's Responses to Questions

**Comments to the Authors:**

Reviewer #1: My comments have been addressed.

Reviewer #2: The authors addressed most of my concerns except the my question #3. It is nice that the authors added a paragraph (lines 276-385) to suggest the use of stringent relatedness score in family data. However, it is still not clear how the relatedness score will affect the result and how to choose the relatedness score in the real data analysis. It is better to test the method using a variety of relatedness score.

**Have all data underlying the figures and results presented in the manuscript been provided?**

Reviewer #1: None

Reviewer #2: Yes

PLOS authors have the option to publish the peer review history of their article (what does this mean?). If published, this will include your full peer review and any attached files.

Reviewer #1: No

Reviewer #2: No
---

## [Decision Letter · Decision Letter 2]

28 Feb 2020

Dear Liu,

We are pleased to inform you that your manuscript 'DRAMS: A Tool to Detect and Re-Align Mixed-up Samples for Integrative Studies of Multi-omics Data' has been provisionally accepted for publication in PLOS Computational Biology.

Best regards,

Ilya Ioshikhes

Associate Editor

PLOS Computational Biology

Thomas Lengauer

Methods Editor

PLOS Computational Biology

Reviewer's Responses to Questions

**Comments to the Authors:**

Reviewer #2: My comments have been addressed.

**Have all data underlying the figures and results presented in the manuscript been provided?**

Reviewer #2: Yes

PLOS authors have the option to publish the peer review history of their article (what does this mean?). If published, this will include your full peer review and any attached files.

Reviewer #2: No

---

## [Editor Report · Acceptance letter]

24 Mar 2020

PCOMPBIOL-D-19-01762R2 

DRAMS: A Tool to Detect and Re-Align Mixed-up Samples for Integrative Studies of Multi-omics Data

Dear Dr Liu,

I am pleased to inform you that your manuscript has been formally accepted for publication in PLOS Computational Biology. Your manuscript is now with our production department and you will be notified of the publication date in due course.

With kind regards,

Bailey Hanna
